# Implementation of a Pharmacogenomic Testing Service through Community Pharmacy in the Netherlands: Results from an Early Service Evaluation

**DOI:** 10.3390/pharmacy9010038

**Published:** 2021-02-12

**Authors:** Tracey Thornley, Bernard Esquivel, David J. Wright, Hidde van den Dop, Charlotte L. Kirkdale, Essra Youssef

**Affiliations:** 1Boots UK, Thane Road, Nottingham NG90 1BS, UK; charlotte.kirkdale@boots.co.uk; 2School of Pharmacy, University of Nottingham, University Park, Nottingham NG7 2RD, UK; 3OneOme, 807 Broadway St. NE, Suite 100, Minneapolis, MN 55413, USA; bernardesquivel@oneome.com; 4University of East Anglia, Norwich Research Park, Norwich NR4 7TJ, UK; d.j.wright@uea.ac.uk (D.J.W.); e.youssef@uea.ac.uk (E.Y.); 5Alliance Healthcare, De Amert 603, 5462 GH Veghel, The Netherlands; Hidde.van.den.Dop@Alliance-Healthcare.nl

**Keywords:** community pharmacy, implementation, medicines optimization, pharmacogenomics

## Abstract

Community pharmacy services have evolved to include medical and pharmaceutical interventions alongside dispensing. While established pharmacogenomic (PGx) testing is available throughout the Netherlands, this is primarily based in hospital environments and for specialist medicines. The aim of this work was to describe how best to implement PGx services within community pharmacy, considering potential barriers and enablers to service delivery and how to address them. The service was implemented across a selection of community pharmacies in the Netherlands. Data were captured on test outcomes and through a pharmacist survey. Following testing, 17.8% of the clinical samples were recommended to avoid certain medication (based on their current medicines use), and 14.0% to have their dose adjusted. Pre-emptive analysis of genotyped patients showed that the majority (99.2%) had actionable variants. Pharmacists felt confident in their operational knowledge to deliver the service, but less so in applying that knowledge. Delivering the service was believed to improve relationships with other healthcare professionals. These results add to the evidence in understanding how PGx can be delivered effectively within the community pharmacy environment. Training pharmacists in how to respond to patient queries and make clinical recommendations may enhance service provision further.

## 1. Introduction

The use of pharmacogenomics (PGx) to support a personalized medicines approach can help improve patient safety and lead to better outcomes for patients. Not only does PGx provide clinical benefits, but it has also been demonstrated to have a positive impact on health costs, with the potential to be cost-effective or cost-saving [1]. Testing has historically been in specialist areas where there are narrow therapeutic windows with serious clinical consequences as a result of mismatched gene–drug pairings. Although most medicines prescribed in primary care may not be considered high risk, the overall prescription volumes of those that are, in combination with high frequency of actionable phenotypes, results in a high potential global impact. Within the Netherlands, a recent estimation of the impact of pre-emptive PGx testing of a panel of 45 drugs indicated that 23.6% of all new prescriptions were linked to an actionable gene–drug interaction (GDI) [2]. The potential to use PGx testing to tailor medicines usage is a natural fit for community pharmacy, where medicines optimization is a key offering of pharmacists through services such as the New Medicines Service in the United Kingdom (UK) [3].

While PGx testing laboratories have been set up across many countries, the practice is not yet embedded into routine care across community and hospital settings. Attempts are being made to develop its use; for example, the UK has set up a National Genomics Medicine Service with the aim to integrate genomic medicine into routine National Healthcare Service care by 2025 [4]. The Netherlands is one of the most advanced in this area, helped by a healthcare system setup around a single, central drug database (G-Standaard) that provides a supportive infrastructure for national testing programs [5]. In 2005, a specialist laboratory was set up to provide national testing facilities at the University of Rotterdam [6] and the ‘Dutch Pharmacogenetics Working Group’ (DPWG) was formed [7]. The DPWG develop PGx-based therapeutic (dose) recommendations, which they have for over 80 drugs, and is updated every three months [8]. Recommendations appear as clinical decision support alerts whenever a medicine that can be informed by PGx is prescribed or dispensed.

Work is ongoing to establish the optimal model for implementing PGx testing into healthcare systems: whether to use a pre-therapeutic single gene approach (historically used), or pre-emptive panel-based approach; who should lead on the testing; in what setting (community or hospital); and finally, whether to use the DPWG or the US-developed Clinical Pharmacogenetics Implementation Consortium (CPIC) recommendations [9], or a combination of the two. Efforts to harmonize DPWG and CPIC recommendations are underway [10], but further research is needed to explore these other issues, particularly on the practical aspects of delivery.

The first randomized controlled trial to use a panel-based approach, PREemptive Pharmacogenomic testing for prevention of Adverse drug Reactions (PREPARE), is ongoing at multiple sites across Europe [11], but results have not yet been reported. There have been a small number of pilot services evaluating panel-based PGx testing in the community pharmacy setting. The Royal Dutch Pharmacist Association pilot [12] and the Implementation of Pharmacogenetics into Primary care Project (IP3 study) [13,14] were both conducted in the Netherlands and found that between 24% and 31% of tests resulted in action being taken (total 215 and 200 patients respectively, tests were provided free for the patient). In the US, a study across six primary care settings (including one pharmacy) identified 96.8% of the 189 patients had at least one actionable phenotype for medications linked to the decision support software, which was then subsequently used to aid medication decisions 236 times by physicians and pharmacists over a period of three months [15]. Limited experience in community pharmacies in other countries has been reported [16].

To enable healthcare professionals to adopt and implement new services of this nature effectively, it is important that all barriers are identified and addressed, and enablers appropriately used. The aims of this service evaluation were to describe the initial PGx outcomes from the service, the proportion of test results which could be used to inform future prescribing and identify whether potential barriers and enablers had been appropriately addressed or used to optimize service delivery from the perspective of the community pharmacist.

## 2. Materials and Methods

### 2.1. Study Design

The study is a service evaluation based on a novel service set up across a network of community pharmacies in the Netherlands between 2019 and 2020. The survey was designed in England with collaborators from the Netherlands, with data managed and quality assured through the use of anonymized data and collation on a centralized database.

### 2.2. Service Implementation

Community pharmacists that were part of a network were invited to take part in the service through local communication channels and networks. Pharmacists across 18 pharmacies attended training during November 2019, and staff across an additional 60 pharmacy sites were trained in February 2020. All pharmacists participating in the service were required to complete the online training provided by KNMP (“Do you already have your DNA passport?” and “Pharmacogenetics: from basics to expertise”) [17], followed by a half day workshop which covered additional information on PGx delivered by a medical expert, covering local guidelines, operational and logistical aspects of the service. Pharmacists engaged with local doctors to raise awareness of the service; offering them the chance to ask any questions, and to arrange methods of communication for any patient results. Local doctors, pharmacists and their healthcare teams were also able to experience the test themselves for free (herein known as educational tests) to enable them to better understand the process and its potential value for patients.

Suitable patients could be identified by pharmacists, referred into the service by local doctors, or were able to request the service themselves within the pharmacy (marketing materials were available within the pharmacy itself). The service took place inside a consultation room within the pharmacy, where pharmacists entered test details such as sample ID, date of birth, gender, ethnicity and medication use (optional) onto the OneOme testing portal. DNA samples were collected from participants using the OraCollect^®^ DNA collection kit (DNA Genotek, Ottawa, Ontario, Canada), and informed consent was obtained from all participants. Samples were sent to OneOme for testing (Minneapolis, MN, USA). The RightMed pharmacogenomic test was run on the samples using TaqMan^®^ SNP Genotyping, (Thermo Fisher Scientific, Waltham, MA, USA) and copy number variation on a qPCR IntelliQube^®^ qPCR platform (Douglas Scientific, Alexandria, MN, USA). All genotyping and PGx interpretation were conducted in a Clinical Laboratory Improvements Amendments (CLIA) and College of American Pathologists (CAP) accredited environment. Each participant received the RightMed Comprehensive test report, which provides PGx interpretation for 27 genes (111 alleles) and guidance on more than 300 medications.

Recommendations from the test outcome were based on local guidelines within the Netherlands [8], alongside a more detailed report which included additional information based on evidence and guidelines from other countries [9]. Results were returned to the pharmacy within one to two weeks via the OneOme portal. Pharmacists then consulted with the patients directly to discuss the outcomes and communicated with the patients’ doctor regarding any potential medication changes. A copy of the summary and full test report was given to the patient directly and uploaded to their medication record; available to healthcare practitioners involved in their care to help inform medicines related decisions.

Anonymized genotype data for genetic variants tested by the OneOme PGx panel were analysed to calculate the frequency at which actionable variants occurred. This was used to estimate potential impact of the OneOme PGx panel on the Dutch primary care population.

### 2.3. Service Evaluation Methods

Volumes of overall tests undertaken were recorded on the OneOme testing portal, which were labelled as either educational (undertaken on doctors and pharmacists) or clinical (undertaken on patients). To support service evaluation and improvement, pharmacists were asked to record additional information onto the OneOme portal when patients returned to the pharmacy after testing with any subsequent prescriptions. This was to allow for enough time for any changes to the prescription to be made following the PGx test. Data were collected on demographics (age, ethnicity, gender), reason for test, test outcome, how the patient accessed the test, why they undertook it, and outcome of the recommendation to the doctor (whether medication was altered and if so how).

Pharmacists participating in the service were invited to complete a survey two to six months after service commencement. The survey was designed to identify barriers and enablers to service implementation from the perspective of the pharmacist. Questions were included to explore feedback on training, motivation for providing the service, preparation for the role, impact on relationships with other healthcare providers, perceived patient benefits, outcomes following the test, and delivery of the service. Responses to survey questions were categorical (yes/no; multiple choice), ordinal (five-point Likert scale) and free-form text. The survey was translated into Dutch before sending electronically to pharmacists.

### 2.4. Data Analysis

Anonymized data were entered onto Microsoft Excel© 365, translated back to English where necessary, and analysed descriptively. Categorical and ordinal data were presented as numbers and percentages as appropriate. The five-point Likert scales were amalgamated to three to simplify data presentation where appropriate.

## 3. Results

### 3.1. Testing Results

From September 2019 to June 2020, a total of 611 tests were undertaken across 22 pharmacies (207 clinical (patients), 404 educational (healthcare professionals)). Demographic and outcomes data were recorded by the pharmacists for 107 patients of the clinical tests (51.7%). While not the main focus of this evaluation, data were also available for 148 (36.6%) of the educational tests. Just over half (56.1%, *n* = 60/107) of the clinical sample were female, with an average age of 59.5 years (range 5 to 87), and 91.6% (*n* = 98/107) were white or Caucasian (remainder unknown). For the educational sample, 68.9% (*n* = 102/148) were female, with an average age of 46.7 years (range 17 to 84), and of the known data, all (*n* = 65/65) were white or Caucasian (remainder unknown *n* = 83).

Over half the patients (52.3%, *n* = 56/107) approached the pharmacist directly to request the test, 24.3% (*n* = 26/107) were recommended by the pharmacist to have the test, and 23.4% (*n* = 25/107) recommended by their doctor. The majority of reasons for requesting the test were due to concern regarding adverse drug reactions or pre-emptive to optimise initial therapy selection (Table 1).

Following testing, 17.8% (*n* = 19/107) of the clinical samples were recommended to avoid certain medication (based on their current medicines use), and 14.0% (*n* = 15/107) to have their dose adjusted (no change for 68.2%, *n* = 73/107). The majority of recommendations were actioned by the prescriber resulting in a change to the patient’s prescription (82.4%, *n* = 28/34).

For the educational results, the majority of the sample recorded the reason for taking the test as other (due to the fact that it was part of attending the training course, or engagement with local healthcare professionals, 80.4%, *n* = 119/148), or as pre-emptive (14.9%, *n* = 22/148). A small number of individuals (4.7%, *n* = 7/148) took the test for clinical reasons. Following testing, 8.8% (*n* = 13/148) of the sample were recommended to avoid certain medication (based on current use), and 4.7% (*n* = 7/148) to have their dose adjusted (no change for the remainder). A third of this sample had their prescriptions altered as a result of the test (30.0%, *n* = 6/20).

The OneOme panel identified one or more actionable variants in 99.2% of the genotyped patients in the sample (*n* = 618). Furthermore, 90.9% of patients had two actionable PGx variants and 57.1% had three actionable PGx variants. Only 1.6% of patients had no actionable variants for *CYP2D6*, *CYP2C9*, *CYP2C19*, *SLCO1B1*, and *VKORC1* genes.

### 3.2. Pharmacist Survey

The survey was completed by 22 pharmacists (managers and/or owners) during June 2020, when they had been providing the service for an average of 27.6 weeks (range 9–64 weeks). Primary reasons for providing the service included being able to personalize patients’ medication (22.7%, *n* = 5), the innovative nature of the service (22.7%, *n* = 5), developing the profession (18.2%, *n* = 4), personal interest (13.6%, *n* = 3), patient need (9.1%, *n* = 2), commercial value (9.1%, *n* = 2), and being able to widen services on offer within that pharmacy (4.5%, *n* = 1).

The majority of pharmacists (86.4%, *n* = 19) agreed that they had sufficient knowledge and background information about PGx following the training and workshops (two disagreed and one was neutral). Pharmacists felt that they had sufficient knowledge in operational aspects of service delivery (introducing the test to patients, taking the swab, and registering details), but less so in applying that knowledge (responding to questions from patients, assessing the report and making recommendations, and discussing results with doctors) (Table 2).

Pharmacists offered free educational tests to doctors as part of early engagement plans, of which a third of pharmacists responding to the survey (31.8%, *n* = 7) said that all the doctors they regularly worked with had used the test, over half (54.5%, *n* = 12) said that some of the doctors had used it, and 13.6% (*n* = 3) said none had. A quarter of pharmacists (27.3%, *n* = 6) felt that doctors supported use of the test with patients, 45.5% (*n* = 10) were neutral, and 27.3% (*n* = 6) stated that they disagreed with its use.

Pharmacists communicated results of the tests with the patients’ doctor via several methods, the most frequent being face to face (45.0%, *n* = 9) or via email (35.0%, *n* = 7). Only three pharmacists (15.0%) were able to update the patient’s medical record directly. Other methods of communication used included via the patient (*n* = 3), and via post (*n* = 2).

Half of pharmacists (54.5%, *n* = 12) stated that offering the service had helped them improve the relationship they have with other healthcare providers, 22.7% (*n* = 5) were not sure, and the same said that it had not (*n* = 5). The majority of pharmacists had received mainly positive feedback from doctors (86.7%, *n* = 19), with the remaining not having any feedback at the time of the survey.

When pharmacists recruited patients directly, uptake of the service varied (nine pharmacists stating 0–50% uptake, nine pharmacists had 51–100% uptake, four unknown); with three-quarters of pharmacists (*n* = 17) stating that the main reason patients did not take up the test was due to the cost.

Over half of the pharmacists (*n* = 14) reported that patients had given them positive feedback about the service (nil responses from the other pharmacists). Pharmacists perceived the main benefits of the service to be around supporting medicines optimization, and in particular more targeted (*n* = 13) and appropriate therapy on drug initiation (*n* = 11) with fewer side effects (*n* = 14).

## 4. Discussion

This evaluation showed that one in six patient tests resulted in recommendations to stop current treatment and one in seven to change the current dose (of which almost all changes were subsequently implemented by the prescriber). This is in line with the higher end of previous studies within the Netherlands, but is understandable given the extended testing panel and evolution of additional evidence over time [12,13,14]. Nearly all patients received a result which could be used to inform future prescribing decisions, and consequently it could be argued that provision of access to such a service is likely to provide benefit to a significant proportion of patients in the future. The value of pre-emptive testing has started to be more widely recognized, with the UK strategy having recently been announced whereby it will be part of routine care attached to medical records to guide therapeutic decision making in the next ten years [18]. Regardless of when, where and why the initial test was carried out, the long-term value to the patient will come from all healthcare professionals using that information to guide any future prescribing decisions.

Pharmacists delivering the service, local doctors, and healthcare teams were offered the test for educational purposes, to enable them to understand the logistics and process for implementation and the potential benefits of the service from a personal perspective. Teams local to the pharmacies were offered the service for purposes of promotion and to encourage referrals but also to enhance social influence, i.e., increase the likelihood of doctors being supportive of the new service. The majority of the educational tests were pre-emptive and not based on clinical need, and as a result, the numbers with recommended changes to medicines were much lower than that seen in the clinical tests.

The training provided pharmacists with sufficient knowledge to be able to operationally deliver the service, although a proportion felt less equipped to be able to respond to questions from patients, assess the report and make recommendations, including discussing these with doctors. The perceptions of local doctors’ response to the provision of PGx was, however, less positive, with only a quarter believing that the majority of doctors were supportive (although half were also perceived as neutral). Interestingly, however, more than half of the pharmacists reported that the service had improved working relationships with local doctors, and most had experienced positive feedback.

Over half of the patients approached the pharmacists directly to enquire about undertaking the test, motivated by specific clinical needs. Despite the high cost in comparison to other services on offer within the pharmacy, patients appeared willing to pay; placing high value on the information to inform choice of therapy. This was also the case when referred directly by the doctor, or by the pharmacist themselves, with high numbers of those recommended the service taking it up, potentially demonstrating trust in the advice from both healthcare professionals. Cost, however, was perceived by pharmacists to be the biggest barrier to uptake by patients who were recommended the service and chose not to participate despite patients being used to paying for access to healthcare (through monthly premiums, deductible fees, and consultations with doctors). These findings are similar to previous research that found that the majority of patients would only undertake testing if reimbursed, despite being interested and valuing the test [19]. This may be more of an issue in countries where healthcare is free at the point of care.

The lack of understanding of PGx has been a commonly reported problem [20], with many healthcare professionals’ perception of the complexity of the application being cited as a barrier to uptake [21]. Healthcare professionals working collaboratively across settings to support patients with their health and medical needs has been found to benefit patient outcomes and deliver value to the healthcare system [22], but use of technology to allow this (namely electronic transfer of health data between settings) has been reported to be of limited interest to health authorities and not widely used among pharmacies [23]. Within the evaluation of this service, most of the pharmacists said that they had received positive feedback from doctors, and it had helped them improve the relationships they had with them more generally, as found in other studies where pre-existing relationships exist [24]. Very few pharmacists had the ability to upload the results of the test directly onto the patients’ health records, which could then be used for future medicine related decisions by all prescribers. The ability for community pharmacists to use and access health records is something that is progressing across many countries [25], allowing more effective communication between healthcare settings. Participating pharmacists were, however, able to follow patients up when they came into the pharmacy for subsequent prescriptions, checking on any changes that were made to their medication, and reinforcing advice and medicines usage. Using pharmacists’ expertise to support medicine optimization provides an opportunity to build on these working relationships more effectively and allows doctors to recognize the value that pharmacists can bring in supporting patient care. Without this partnership approach, the value of the test will not be realized in following through medication recommendations and monitoring changes.

Given the backdrop of widespread PGx services across hospitals in the Netherlands, it is not surprising that the majority of pharmacists in the analysis were interested in getting involved as a way of expanding their professional role in providing innovative services. While these pharmacists perceived PGx as a natural extension to their role in supporting medicines optimization, levels of confidence and knowledge of pharmacists in countries where PGx testing is not established in the setting have been reported to be low [26,27,28].

### Limitations

Due to COVID-19, PGx-related activities within community pharmacies were reduced from March to June 2020, and therefore the number of patients in receipt of the service in each pharmacy per month was relatively low. Additional demographic and outcomes of the service data were collected as part of the evaluation when patients returned to the pharmacy, and hence were not available for all service users. In comparison, the large number of tests undertaken for educational reasons reflects that this evaluation was at an early stage, where community pharmacists and the local doctors were being introduced to the new technology. While education regarding pharmacogenomics is a significant element of pharmacist development within the Netherlands, the selection of community pharmacists may also not be wholly representative of the community at large as they were selected based on their interest in providing the service and may therefore be more proactive than other pharmacists. A later evaluation may provide a more accurate picture with respect to demand and the consequences of any prescription modifications (treatment effect, etc.). The survey was not piloted or validated, and the questions were developed by the team who made assumptions as to what the barriers were likely to be and therefore may not have captured all individual or environmental elements related to service implementation or effectiveness. Further qualitative work is warranted to better understand the barriers and enablers to service provision from the perspective of all stakeholders (with patient and doctor perceptions being indirectly reported through pharmacists’ responses). Pharmacists were responsible for only a quarter of all tests, and although the service was delivered through them, research to identify barriers and enablers associated with direct patient access and physician referral is warranted. Like all healthcare services, the impact of COVID-19 affected our ability to follow up with additional patients when they presented in the pharmacy, thereby limiting the number of longitudinal data points.

## 5. Conclusions

Each country looking to implement PGx testing will have a different set of criteria for patient and drug eligibility based on health economic value to healthcare funders, and for where the test itself is conducted (be it within community of hospital environments). The importance of ensuring that regardless of why and where the test was conducted, the results should be used by all healthcare professionals and across all settings going forward to support optimum medicines use, providing longer term value to the patient and healthcare system.

## Figures and Tables

**Table 1 pharmacy-09-00038-t001:** Reason for test (clinical sample, *n* = 107).

Reason for Test	Count	Percentage
Previous/current adverse drug event	29	27.1%
Pre-emptive	25	23.4%
Other	17	15.9%
Psychiatric condition	14	13.1%
Cardiovascular condition	12	11.2%
Polypharmacy	5	3.7%
High-risk medication (s)	4	3.7%
Oncologic condition	1	0.9%
Total	107	100%

**Table 2 pharmacy-09-00038-t002:** Level of pharmacists’ agreement with having sufficient knowledge following the training (*n* = 22).

Area of Knowledge	Agreed/Strongly Agreed	Neutral	Disagreed
Introduce the PGx test to patients	18 (81.8%)	3 (13.6%)	1 (4.5%)
Respond to concerns and/or questions from patients	12 (54.5%)	8 (36.4%)	1 (4.5%)
Take the swab	21 (95.5%)	1 (4.5%)	0 (0.0%)
Register and send the swab online	19 (86.4%)	2 (9.1%)	1 (4.5%)
Assess report and make recommendations	10 (45.5%)	9 (40.9%)	3 (13.6%)
Discuss the results with a doctor	15 (68.2%)	4 (18.2%)	3 (13.6%)

## Data Availability

Data can be made available upon request to the corresponding author.

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
