# Peer review of "Implementation of a Pharmacogenomic Testing Service through Community Pharmacy in the Netherlands: Results from an Early Service Evaluation"

_pharmacy, 2021, doi:10.3390/pharmacy9010038_

Round 1
Reviewer 1 Report
The authors describe the introduction of a new pharmaceutical service to community pharmacies.
Please separate the conclusions from the discussion.
Author Response
Th
The authors describe the introduction of a new pharmaceutical service to community pharmacies.
Please separate the conclusions from the discussion.
Text from line 313 in the discussion has been split out to a new conclusion section in line 346.
Reviewer 2 Report
Thornley et al implemented PGx test in a selection of community pharmacies in Netherlands, and studies how PGx test results affect medication prescription. It showed 99.2% patients carry actionable variants. These results demonstrated the importance of PGx delivering and training through community pharmacy.
- The way of community pharmacies were selected may be biased. For example, the community selected have higher PGx education level so they are more proactive than other communities. Thus the evaluation results may not represent the overall situation in the Netherlands.
- The consequence of prescription modification led by PGx test is also important. Did patient reach better treatment effect? Or avoid certain side effects? This would be an important index in evaluating the impact of community PGx test.
Author Response
Thornley et al implemented PGx test in a selection of community pharmacies in Netherlands, and studies how PGx test results affect medication prescription. It showed 99.2% patients carry actionable variants. These results demonstrated the importance of PGx delivering and training through community pharmacy.
The way of community pharmacies were selected may be biased. For example, the community selected have higher PGx education level so they are more proactive than other communities. Thus the evaluation results may not represent the overall situation in the Netherlands.
We have added this as a limitation in the discussion section (Line 328: “Whilst education regarding pharmacogenomics is a significant element of pharmacist development within the Netherlands, the selection of community pharmacists may also not be wholly representative of the community at large as they were selected based on their interest in providing the service and may therefore be more proactive than other pharmacists.”)
The consequence of prescription modification led by PGx test is also important. Did patient reach better treatment effect? Or avoid certain side effects? This would be an important index in evaluating the impact of community PGx test.
Unfortunately, we were not able to track these outcomes for patients as it was beyond the scope of the service and evaluation. We have also added this to the limitations section (Line 333 (bold text): “A later evaluation may provide a more accurate picture with respect to demand and consequences of any prescription modifications (treatment effect etc).”)
Reviewer 3 Report
In general, the article is well written, but to be more solid, the authors should consider the following points:
- Introduction: the rationale of the study should be mentioned in detail. It would be helpful to have more information on how this study finding might be applied to implementing pharmacogenomic testing service through community pharmacy.
- Methods: methods are not sufficiently described with regards to data collection tools and their validation, data analysis and interpretation, data quality control, and management, study design, and settings. Therefore, authors should address these points clearly in the methods.
- There is no subheading regarding statistical analysis; it would be better in the methods?
- As for data collection tool validation, nothing is mentioned because the authors prepared the tool, which should be validated. Hence, it should be elaborated in the discussion it was not validated. Besides, authors should mention how many points of the Likert scale they used and how it was categorized and analyzed regarding the Likert scale. As I have seen in the result, they use four points Likert scale, and the participant responses are categorized into three, but it was not mentioned in the methods how it would be analyzed. The responses of "agree" and "strongly agree" were reported together. Why? If there is a "strongly agree" option, there should also be a "strongly disagree" choice. Hence, authors should clearly mention this matter in the methods.
- As for implementation, the recently published study would be helpful and I suggest you take a look at this study “Baldoni S, Pallotta G, Traini E, Sagaro GG, Nittari G, Amenta F. A survey on feasibility of telehealth services among young Italian pharmacists. Pharm Pract (Granada). 2020 Jul-Sep;18(3):1926. doi: 10.18549/PharmPract.2020.3.1926. Epub 2020 Aug 6. PMID: 32802217; PMCID: PMC7416313. ”
- The quality of writing is good, but there are few typos and grammar errors and to be corrected.
Author Response
In general, the article is well written, but to be more solid, the authors should consider the following points:
Introduction: the rationale of the study should be mentioned in detail. It would be helpful to have more information on how this study finding might be applied to implementing pharmacogenomic testing service through community pharmacy.
We believe the rationale is provided within the final sentence where we state that the evaluation is to identify whether potential barriers or enablers have been appropriately addressed or utilised to optimise service delivery. As this evaluation is of a community pharmacy based service we have however changed ‘pharmacist’ to ‘community pharmacist’ at the end of the Introduction. (Line 88).
Methods: methods are not sufficiently described with regards to data collection tools and their validation, data analysis and interpretation, data quality control, and management, study design, and settings. Therefore, authors should address these points clearly in the methods. There is no subheading regarding statistical analysis; it would be better in the methods?
We have added in brief study design (Line 91–94: “The study is a service evaluation based on a novel service set up across a network of community pharmacies in the Netherlands between 2019 and 2020. The survey was designed in England with collaborators from the Netherlands, with data managed and quality assured through the use of anonymized data and collation on a centralised database”) and data analysis sections (Lines 160–164: “Anonymized data were entered onto Microsoft Excel© 365, translated back to English where necessary and analyzed descriptively. Categorical and ordinal data were presented as numbers and percentages as appropriate. The five-point Likert scales were amalgamated to three to simplify data presentation where appropriate.”)
As for data collection tool validation, nothing is mentioned because the authors prepared the tool, which should be validated. Hence, it should be elaborated in the discussion it was not validated.
The limitations section has been amended to make it clear that the survey was not validated (Line 334)
Besides, authors should mention how many points of the Likert scale they used and how it was categorized and analyzed regarding the Likert scale. As I have seen in the result, they use four points Likert scale, and the participant responses are categorized into three, but it was not mentioned in the methods how it would be analyzed. The responses of "agree" and "strongly agree" were reported together. Why? If there is a "strongly agree" option, there should also be a "strongly disagree" choice. Hence, authors should clearly mention this matter in the methods.
Line 156 already states the number of points in the Likert scale, but we have added Line 164: “The five-point Likert scales were amalgamated to three to simplify data presentation where appropriate.” in addition.
As for implementation, the recently published study would be helpful and I suggest you take a look at this study “Baldoni S, Pallotta G, Traini E, Sagaro GG, Nittari G, Amenta F. A survey on feasibility of telehealth services among young Italian pharmacists. Pharm Pract (Granada). 2020 Jul-Sep;18(3):1926. doi: 10.18549/PharmPract.2020.3.1926. Epub 2020 Aug 6. PMID: 32802217; PMCID: PMC7416313. ”
We have reviewed the suggested paper Baldoni et al, and added text at Line 289 “but use of technology to allow this (namely electronic transfer of health data between settings) has been reported to be of limited interest to health authorities and not widely used among pharmacies [23]”.
The quality of writing is good, but there are few typos and grammar errors and to be corrected.
We have checked through again (e.g. Line 33) pharmacogenomics corrected.
Reviewer 4 Report
The authors present an interesting work showing how important it is to introduce new pharmacogenetic tests into practice. A detailed analysis was carried out. The article is written in too much detail. It is necessary to reduce the content of the chapters methods and results, concentrating the main emphasis on practical results that are significant for the future.
Author Response
The authors present an interesting work showing how important it is to introduce new pharmacogenetic tests into practice. A detailed analysis was carried out.
The article is written in too much detail. It is necessary to reduce the content of the chapters methods and results, concentrating the main emphasis on practical results that are significant for the future.
We feel that as this is a service evaluation, and the methods are very important for others who may be thinking about implementing something similar, to learn from, we would prefer not to cut it down; however, we have created a data analysis section which has removed duplication within the methods. Reviewer 3 also asked for additional detail in the methods. We also feel that the results are a full description of all the salient points and would prefer not to cut them down.